



# Machine learning based virtual load sensors for mooring lines using motion and lidar measurements

Moritz Gräfe[1], Vasilis Pettas[1], Nikolay Dimitrov[2], and Po Wen Cheng[1]

[1]University of Stuttgart, Stuttgart Wind Energy (SWE), Allmandring 5b, 70569 Stuttgart, Germany
[2]DTU Wind and Energy Systems, Technical University of Denmark, Frederiksborgvej 399, Roskilde 4000, Denmark
**Correspondence:** Moritz Gräfe (graefe@ifb.uni-stuttgart.de)

**Abstract.** Floating offshore wind turbines (FOWT) are equipped with various sensors that provide valuable data for turbine monitoring and control. Due to technical and operational challenges, load estimations for mooring lines and fairleads can be difficult and expensive to obtain accurately. This research delves into a methodology where floater motion measurements and wind speed measurements, derived from forward-looking nacelle-based lidar, are utilized as inputs for different types of neural networks to estimate fairlead tension time series and damage equivalent loads (DELs). Mooring line loads are intrinsically linked to the dynamics and the position of the floater. Therefore, we systematically analyze the individual contribution of floater dynamics on the prediction quality of fairlead tension time series and DELs. Wind speed measurements obtained via nacelle-based lidar on floating offshore wind turbines are influenced inherently by the platform's dynamics, notably the rotational pitch displacement and surge displacement of the floater. Consequently, the lidar wind speed data indirectly contains the dynamic behavior of the floater, which, in turn, governs the fairlead loads. This study leverages lidar-measured Line of Sight (LOS) wind speeds to estimate mooring line tensions. Training data for the model is generated by the aero-elastic wind turbine simulation tool, openFAST, in conjunction with the numerical lidar simulation framework ViConDAR. The fairlead tension time series are predicted using long-short-term-memory (LSTM) networks. DEL predictions are made using three different approaches. First, DELs are calculated from predicted time series; second, DELs are predicted using a sequence-to-one LSTM architecture, and third, DELs are predicted using a convolutional neural network architecture. Results indicate that fairlead tension time series and DELs can be accurately estimated from floater motion time series. Further, we found that lidar LOS measurements do not improve time series or DEL predictions if motion measurements are available. However, using lidar measurements as model inputs for DEL predictions leads to similar accuracies as using displacement measurements of the floater.

## 1 Introduction

As countries worldwide set ambitious targets for FOWT installations and with numerous upcoming projects in the pipeline, the installed capacity of FOWTs is anticipated to grow significantly in the current decade. Projections indicate that the global installed capacity of floating wind power will increase to 16.5 GW by 2030 GWEC 2022. For the large-scale commercial installation of FOWTs, reliable load-monitoring systems are needed. FOWTs are outfitted with sensors that deliver crucial data for controlling and monitoring single turbines and wind farms. Virtual sensors present a valuable alternative when measuring





specific physical quantities is either challenging or expensive. Specifically, the mooring lines of FOWTs, which are vital for stability, are susceptible to mechanical failures that could lead to severe consequences. Fatigue and extreme loads are key contributors to such failures, particularly in mooring lines and fairleads, as identified by Shafiee (2022). Consequently, precise estimations of loads on these components are essential for effectively monitoring floating offshore wind turbines.

Although mooring lines perform critical functions, monitoring and assessing individual lines' remaining lifetime is difficult.
Different authors (see e.g. Gao and Moan (2007); Benasciutti and Tovo (2005)) have proposed methods to derive fatigue damage of mooring lines in the frequency domain while avoiding the need for rainflow-counting algorithms in the time domain. In principle, these methods could be used in operational settings assuming that representative spectral properties of the target loads are known. However, the harsh marine environment and logistical challenges make installing sensors and maintaining their accuracy difficult over long periods. Studies have investigated various approaches to model and predict mooring line
loads in this context. An overview of modeling approaches for mooring line loads based on physical principles can be found in Borg et al. (2014). Physical mooring line models describe the mooring line behavior using knowledge about its mechanics and underlying physical laws. Different model fidelities, ranging from quasi-static to multi-body and finite element models are described in the literature. While finite element models can calculate mooring line loads very accurately, the high computational costs make them impractical for monitoring applications. Multi-body models, as described in Hall (2020), combine good
accuracy with acceptable computational costs, making them applicable for general monitoring applications. However, the floater dynamics in 6 DoF must be known, which might not be the case in operational settings.

Data-driven models can make predictions without a physical description between input and output quantities at low computational costs. However, training data-driven models typically involves large sets of training data that must be available through measurements or representative simulation models. Different studies have demonstrated the potential of data-driven models
for predicting physical quantities, not only for mooring line load predictions. In the study by Azzam et al. (2021), observable predictor signals were employed to train neural networks for predicting the load on a wind turbine gearbox. Likewise, Dimitrov and Göçmen (2022) explored various network architectures to predict and forecast blade root bending moments, detect wake centers, and assess blade tip-to-tower clearance. In Hlaing et al. (2023), Bayesian neural networks are used for a fleet monitoring approach, where models are trained on measured load data from one fully monitored turbine. At the same time, predictions
are made for other turbines in the fleet. The study uses ten-minute statistics of different available time series measurements as input and predicts damage equivalent tower bending moments. Similarly, de N Santos et al. (2023) used a fleet monitoring approach to predict long-term damage accumulation of tower bending moments employing physics-informed neural networks.

Walker et al. (2022) developed data-driven digital twin models for predicting and forecasting mooring line tension based on measured operational time series data, including floater dynamics signals and ambient wind speed. The developed model could
predict mooring line tension time series accurately and make near future (1 minute) tension forecasts. The use of predicted time series for DEL analysis was not investigated. In contrast to this work, our study is focused on leveraging lidar inflow measurements for mooring line load predictions and predicting DELs by utilizing different modeling approaches.

Various studies have used Nacelle-based lidar inflow measurements for turbine load calculation. For instance, in Dimitrov et al. (2019) and Conti et al. (2021), besides lidar estimated wind field statistics for parameterization of constrained wind fields,





measured lidar wind speed time series are used to constrain synthetic turbulent wind fields. These wind fields are further used to simulate turbine responses using an aero-elastic simulation code. While the method yields good results for load validation, it is not suitable for real-time monitoring applications. In our study, we aim to demonstrate the ability of data-driven lidar-based models to predict mooring line tension in real-time. The idea of this approach was first presented in Gräfe et al. (2023). This initial study demonstrated the general feasibility but did not analyze all relevant aspects systematically. We build on the existing study while adding different aspects to the present study. The feature selection is made systematically, allowing conclusions on the contribution of individual input features for the prediction quality. To understand the effect of different lidar patterns better, we investigate four different lidar patterns with increasing density of focus points and compare the resulting performance. A new approach for the prediction of DELs is introduced. Instead of calculating DELs from predicted time series, we investigate machine learning methods to predict DELs directly from input time series signals. The environmental conditions are sampled considering randomized atmospheric and oceanic conditions in an attempt better to reflect the stochastic nature of real met-ocean conditions. The influence of sensor noise in the models' input features has been considered and evaluated in a comparative study.

## 1.1 Objectives

With the present work, we aim to provide insights into the prediction of fairlead tension time series and DELs using machine learning techniques. Further, we investigate the value of lidar inflow measurements in this context. In short, the objectives of this work are:

- to investigate diverse machine learning approaches suitable for constructing virtual sensor models to predict fairlead tension time series and DELs

- to analyze the contribution of individual input features to the models' performance. Considered input features include platform dynamics and position measurements, SCADA signals, and lidar inflow measurements.

- to evaluate the influence of noise in platform position and dynamics input signals on the model prediction accuracy.

- to investigate the value of lidar inflow measurements for fairlead tension prediction and analyze the sensitivity to different lidar patterns.

## 1.2 Structure of the work

In section 2, the overall methodology of the study, the simulation environment, and the different prediction models are introduced. In section 3 we introduce the case study used to demonstrate the performance of our approach. This section describes the numerical FOWT model, the different investigated lidar configurations, and the environmental conditions for creating the training and testing database. In section 4 we present the results of the time series prediction and DEL prediction models and evaluate their performance. In section 5, a final discussion on the results and the transferability of the proposed models to operational environments is given.



## 2 Methodology

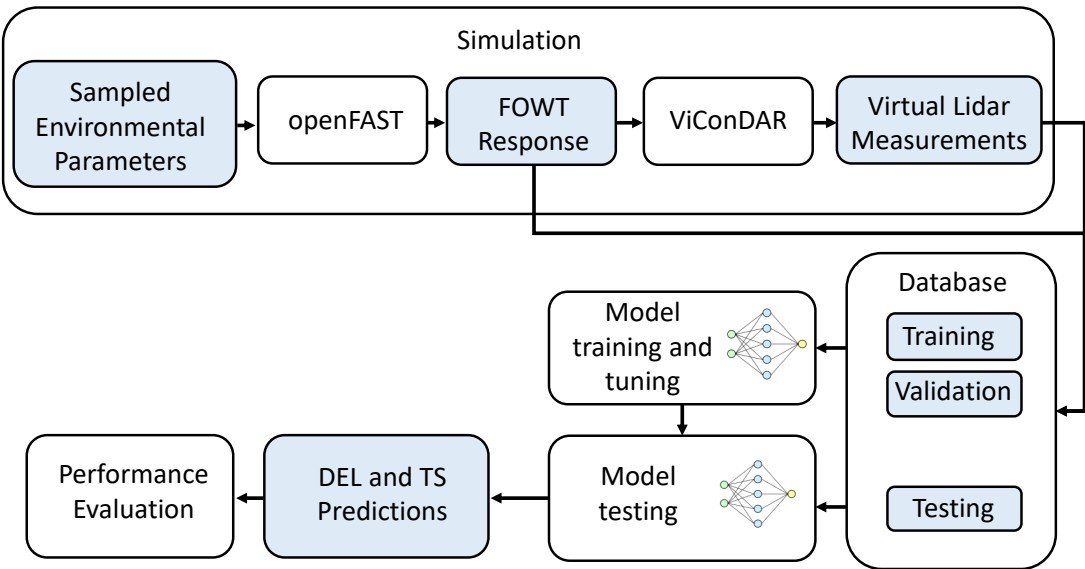

**Figure 1.** Overview of the simulation, training and evaluation methodology. White boxes denote used tools and process steps, blue boxes denote used and generated data sets.

We adopt a numerical approach to create training data for the models and to assess their predictive capabilities. This approach integrates the use of openFAST, an aero-elastic wind turbine simulation code (Jonkman (2007)), with ViConDAR, a numerical lidar simulation code (Pettas et al. (2020); Gräfe et al. (2022)), for generating the necessary data sets. Synthetic turbulence boxes are created employing TurbSim, an open-source wind field generation tool (Jonkman (2014)). We engage openFAST version 3.2.1 to conduct aero-elastic simulations of relevant FOWT responses, including mooring line loads. The turbine responses and synthetic turbulent wind fields, feed into the numerical lidar simulator to produce synthetic lidar data reflecting the influence of motion in six degrees of freedom. The fairlead tensions, turbine dynamics, and turbine responses characterized by blade pitch, power output, and nacelle yaw angles, along with corresponding lidar measurements, are used to train neural networks for predicting fairlead tension time series and DELs. The performance of these predictions is then validated against the generated simulation data. An illustrative diagram outlining this methodology is presented in figure 1.

### 2.1 Numerical lidar simulation environment

Floater movements influence the measurements obtained by nacelle-based lidar systems on FOWTs. A recent study by (Gräfe et al. (2022)) highlights that the motion of the floater-tower assembly, particularly its rotational movements, alters the orientation of the lidar beams. This deviation from fixed systems leads to variations in the LOS velocity measurements and displaces





the lidar's focal points. When a turbulent wind field is probed, this leads to motion-induced variations of the measurements. Furthermore, the floater's combined translational and rotational dynamics introduce velocities to the nacelle. These translational velocities are superimposed on the measurements. The comprehensive analysis and quantification of these influences are detailed in Gräfe et al. (2023). Our research uses these motion-induced variations in lidar data, employing uncorrected LOS velocity readings as inputs for our load prediction models. It should be noted, that the sampling frequency of the lidar limits the effect of platform dynamics on lidar radial wind speed measurements. High-frequency dynamics, higher than half of the lidar sampling frequency cannot be captured. Motion-influenced lidar measurements are simulated using the open-source lidar simulation environment ViConDAR. ViConDAR is a numerical framework for simulating lidar measurements in turbulent wind fields and using simulated measurements as constraints in synthetic wind field generation. ViConDAR has been adapted for consideration of floating dynamics of the lidar system in 6 DOF. To reflect the influence of floater motions it requires the input of the rotational displacements (yaw, pitch, roll), the translational displacements in surge, sway, and heave direction, and the translational velocities in the surge, sway, and heave direction, which are obtained through the openFAST simulations.

## 2.2 Mooring line load determining parameters

Catenary mooring lines, governed by the principles of the catenary shape, are subject to dynamic loading that fluctuates with the motions of the platform. In the case of catenary mooring lines, the forces exerted at the fairlead — both vertical and horizontal — can be collectively described as the fairlead tension force, which is subject to frequent fluctuations, as detailed in studies like those by Hall and Goupee (2015) and Özinan et al. (2020).

Fairlead tensions are intrinsically linked to the dynamics of the floater, which operates in six degrees of freedom. The floater's translational and rotational movements dictate the fairlead's relative positioning to the anchor point. Therefore, these displacement factors primarily govern the quasi-static loading experienced by a catenary mooring line. Moreover, additional loading components on the mooring line arise due to the translational velocities of the fairlead about the anchor. The efficiency of a lidar-based model for predicting fairlead tensions hinges on its ability to accurately interpret turbine displacement and velocity patterns from the measured lidar radial wind speed data. The performance of such a model relies on its capacity to discern these patterns, which are indicative of the underlying dynamics influencing the fairlead tensions.

## 2.3 Virtual Load sensor models

In the context of wind turbine operation, control, and monitoring, the measurement of key physical parameters is crucial. Nonetheless, there are scenarios where direct measurement through conventional sensors is impractical or unattainable. In such instances, the use of virtual sensors presents a viable alternative. These sensors perform indirect estimations of the target quantities by utilizing available measurement data as inputs to model the desired outputs. The nature of the model employed in this process varies based on the understanding of the input-output relationship. When this relationship is clearly defined, physical models can be utilized. Conversely, in situations where the relationship is either unknown or too complex for a physical model to encapsulate, data-driven models can be employed. An example of this is the prediction of fairlead tensions from platform positions, dynamics, and lidar inflow measurements.





This study delves into two distinct applications of the virtual load sensor model. We employ a sequence-to-sequence model for predicting the time series of fairlead tension load. Additionally, we examine and compare three different modeling approaches for predicting fairlead tension DELs.

### 2.3.1 Time series prediction model

The virtual load sensor time series model is built as a sequence-to-sequence LSTM model where the model uses input time series sequences to predict target time series sequences. LSTM networks are recurrent artificial neural networks that have proven particularly effective in capturing time dependencies in input data (Hochreiter and Schmidhuber (1997)). The key feature of LSTMs is their ability to remember or forget previous inputs selectively over extended periods. This is achieved through specialized memory cells that store information over multiple time steps. Unlike traditional neural networks, LSTMs have a gating mechanism that regulates the flow of information between cells, allowing them to selectively retain or discard information based on its relevance to the current state of the network.

The network structure that has been chosen for this model consists of six layers as shown in figure 2. The window size defines the number of timesteps in the input features. Predictions are made for the same number of target time series time steps, while no forecasts into the future are made. A hyperparameter tuning process determines the window size used in this study. Since we further use the predicted time series for calculations of DELs an overall sequence length of 600s is needed. Therefore, input sequences are divided into several shorter sequences based on the optimized window size in the prediction step. Predictions are then made in succession while remembering the lstm cell states for each sub-sequence. Sub-sequences are concatenated after the prediction for the calculation of DELs.

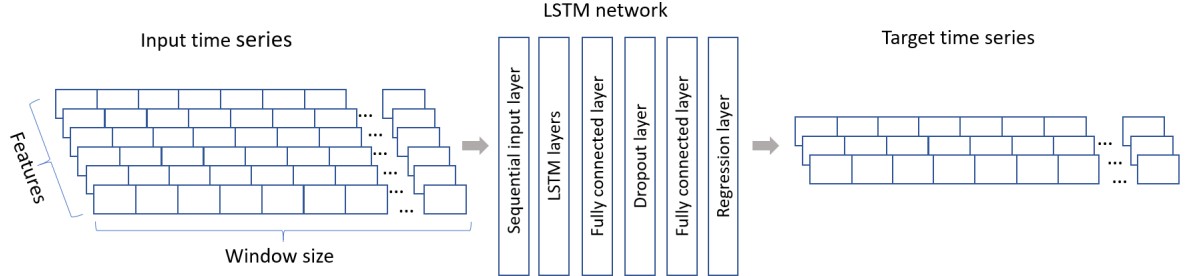

**Figure 2.** Sequence-to-Sequence time series prediction model architecture

### 2.3.2 DEL prediction models

Accurately predicting DELs is vital when considering the potential use and application of virtual load sensors for fairlead tensions. DELs are used to monitor fatigue loads and estimate remaining lifetimes. A rainflow counting algorithm counts the number of cycles $n_i$ with load amplitudes $S_i$ in each sequence. DELs are calculated based on Miner's rule for damage accumulation (Miner (1945)). For each sample sequence $DEL_i$ is calculated by:



$$DEL_i = \sqrt[m]{\frac{1}{N_{\text{ref}}} \sum_i n_i S_i^m} \tag{1}$$

In this equation, $N_{ref}$ represents the reference cycle count, and $m$ denotes the Wöhler exponent. Using the guidelines of DNVGL-OS-E301, the Wöhler exponent for studless mooring chains made of steel is set at $m = 3$. Additionally, the reference
cycle count $N_{ref}$, assuming a reference cycle frequency of 1 HZ is set at 600.

We evaluate the prediction quality of fairlead tension DELs for three different modelling approaches. First, the time series model as described in 2.3.1 is trained to predict fairlead tension time series. The fairlead DELs are then calculated from the predicted and concatenated time series using equation 1. Second, a sequence-to-one regression model architecture predicts DELs directly from the input time series sequences. In this case, the model is trained to predict a single estimate of DEL for
each set of input time series. In contrast to the sequence-to-sequence model for time series prediction the sequence-to-one architecture uses only the last output of the LSTM layers for a given input sequence to produce the DEL prediction. This LSTM output is then passed through a fully connected layer and a regression layer to produce the final DEL estimate for the given input sequence. The model architecture is shown in figure 3.

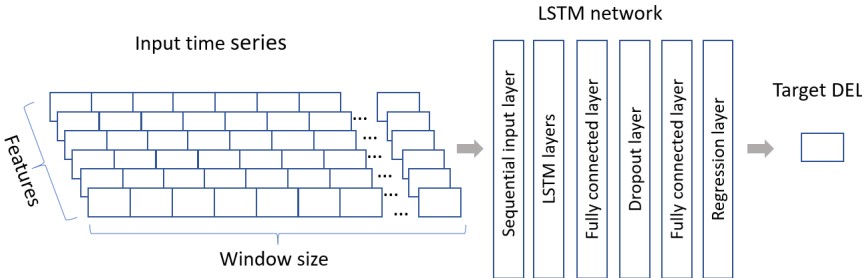

**Figure 3.** Seq2One model architecture

The third investigated DEL prediction approach employs a convolutional neural network. Convolutional Neural Networks
(CNNs) have traditionally been associated with image processing tasks, but they have also proven to be effective for time series classification. CNNs can extract meaningful features from sequential data by using 1D convolutional layers to scan through the time series, identifying patterns within the data. These learned features can capture essential temporal information, such as trends and frequencies, making CNNs useful in applications like the prediction of DELs. The architecture of the model employed in this study is shown in figure 4. The input time series are fed through a one-dimensional convolutional layer,
which applies filters to the input time series, allowing the network to detect patterns and relevant features at various scales and positions within the data. The pooling layer follows the convolutional layers and reduces the spatial dimensionality of the feature maps. The fully connected layer connects all the neurons from the previous layer to each neuron in itself. Finally, the regression layer consists of a single neuron, which produces the final regression output for DEL estimation.

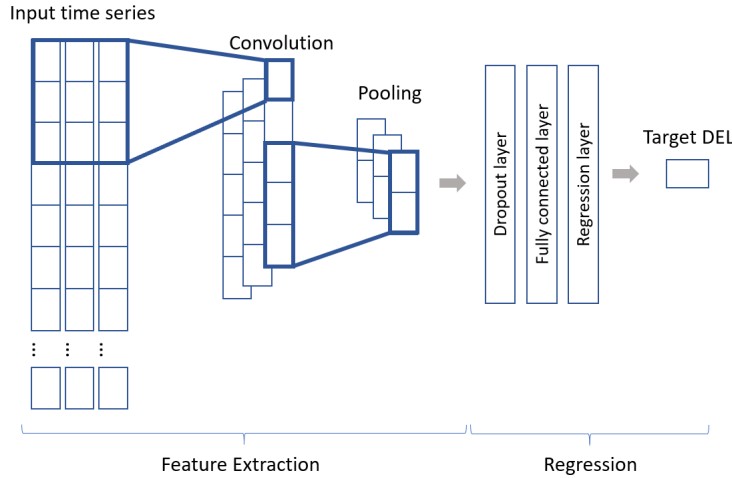

**Figure 4.** One-dimensional convolutional neural network architecture for time series regression architecture

## 2.4 Model performance

The performance of the time series prediction model in this study is assessed using the RMSEN (Root Mean Squared Error Normalized) metric. RMSEN is defined as the root mean squared error between the predicted time series $Y_{pred}$ and the actual target time series $Y$, normalized by the standard deviation $\sigma_y$ of the reference time series. RMSEN is calculated using equation 2.

$$RMSEN = \frac{1}{\sigma_y}\sqrt{\frac{\sum_{i=1}^{N}(Y_{pred}-Y)^2}{N}} \tag{2}$$

In this equation, $Y_{pred}$ represents the predicted values, $Y$ denotes the actual reference values, and $N$ is the total number of data points in the sequence. For a comprehensive evaluation across the entire testing dataset, both $Y_{pred}$ and $Y$ are concatenated, encompassing all test sequences. In this context, $N$ refers to the total count of data points across the complete testing dataset, and $\sigma_y$ is the standard deviation calculated over the entire reference dataset.

DEL predictions are evaluated using absolute percentage errors which allow an easier interpretation of results. Absolute
percentage error between predicted and reference DELs are calculated by:

$$APE = \left|\frac{DEL_{pred}-DEL_{ref}}{DEL_{ref}}\right| * 100 \tag{3}$$

where $DEL_{ref}$ is the reference DEL and $DEL_{pred}$ is the predicted DEL value.

## 2.5 Feature selection

In this study, the selection of input features for various models was carried out manually, aligning with the study's objective
to assess the impact of individual input features on prediction accuracy. Besides the lidar system, we consider input features,



typically available through FOWT monitoring systems in real-world scenarios, including SCADA and IMU/GNSS data. Consequently, scenarios combining platform dynamics, SCADA signals, and lidar measurements were distinctly categorized. As outlined in section 2.2, the platform dynamics across six degrees of freedom (DOF) are crucial in determining fairlead tensions, leading to the inclusion of both platform displacements and velocities in our analysis. Additionally, standard SCADA system

outputs like blade pitch, power, and nacelle yaw signals were considered, given their potential relevance to the prediction task. Lidar Line-of-Sight (LOS) signals were also incorporated for each case to ascertain their specific contribution to prediction accuracy.

To manage the scope of our investigation, platform dynamics were grouped into four categories: translational displacements, rotational displacements, translational velocities, and rotational velocities, with each category encompassing signals in the x,

y, and z coordinates. For cases $A$ to $E$, the model inputs were restricted to platform dynamics, SCADA signals, and nacelle yaw position. In contrast, cases $A_{Lidar}$ to $E_{Lidar}$ were expanded to include radial wind speed measurements from the lidar, facilitating a focused analysis of lidar signals' impact. Case $F_{Lidar}$ is unique, using only the lidar LOS wind speeds and nacelle yaw position, while the nacelle is always perfectly aligned with the wind direction; the latter is included to inform the model about the global wind direction, a parameter not evident from the radial wind speeds. This inclusion is crucial as it informs

the model of the rotor's alignment, subsequently influencing the rotor thrust forces and the resulting floater dynamics. This particular case is designed to evaluate the potential of a model based solely on lidar data in predicting fairlead tensions. A summary of all these scenarios is presented in table 1.

**Table 1.** Summary of all modeling cases and input feaures.

| Feature | A | B | C | D | E | $A_{Lidar}$ | $B_{Lidar}$ | $C_{Lidar}$ | $D_{Lidar}$ | $E_{Lidar}$ | $F_{Lidar}$ |
|---|---|---|---|---|---|---|---|---|---|---|---|
| LOS velocities [ms$^{-1}$] | x | x | x | x | x | ✓ | ✓ | ✓ | ✓ | ✓ | ✓ |
| Nacelle Yaw [deg] | ✓ | ✓ | ✓ | ✓ | ✓ | ✓ | ✓ | ✓ | ✓ | ✓ | ✓ |
| Blade Pitch [deg], Power [W] [°] | ✓ | ✓ | ✓ | ✓ | ✓ | ✓ | ✓ | ✓ | ✓ | ✓ | x |
| Platform transl. displ. (surge, sway, heave) [m] | ✓ | ✓ | ✓ | ✓ | x | ✓ | ✓ | ✓ | ✓ | x | x |
| Platform rot. displ. (pitch, roll, yaw) [deg] | ✓ | ✓ | ✓ | x | x | ✓ | ✓ | ✓ | x | x | x |
| Platform transl. velocities (x, y, z) [m/s] | ✓ | ✓ | x | x | x | ✓ | ✓ | x | x | x | x |
| Platform rot. velocities (x, y, z) [deg/s] | ✓ | x | x | x | x | ✓ | x | x | x | x | x |

## 2.6 Hyperparameter tuning

To enhance model performance, hyperparameters are optimized through a tuning process, similar to the methodology described

by Dimitrov and Göçmen (2022). This optimization differentiates between two categories of hyperparameters. The first category encompasses parameters that shape the architecture and training attributes of the models, such as the number of layers



**Table 2.** Prediction models hyperparameter

| Model | Hyperparameter | Parameter space | case $A$ | case $A_{Lidar}$ |
|---|---|---|---|---|
| TS | Number of LSTM layers | 1-3 | 1 | 1 |
| | Number of LSTM units per layer | 50-200 | 130 | 100 |
| | Number of fully connected layers | 1-3 | 1 | 1 |
| | Window length | 50:50:600 | 100 | 100 |
| | Number of units in fully connected layers | 50-200 | 190 | 60 |
| | Dropout rate | 0-0.5 | 0.25 | 0.3 |
| | Number of training sequences | 200-1800 | 1800 | 1800 |
| DEL Seq2One | Number of LSTM layers | 1-3 | 1 | 1 |
| | Number of LSTM units per layer | 50-200 | 130 | 110 |
| | Number of fully connected layers 1 | 1-3 | 1 | 1 |
| | Number of units in fully connected layers | 50-200 | 80 | 150 |
| | Dropout rate | 0-0.5 | 0.3 | 0.26 |
| | Number of training sequences | 200-1800 | 1800 | 1800 |
| DEL Convolution | Number of Convolutions | 1-3 | 2 | 2 |
| | Filter size | 1-30 | 25 | 17 |
| | Number of filters | 1-100 | 55 | 67 |
| | Pool size | 1-50 | 6 | 1 |
| | Dropout rate | 0-0.5 | 0.2 | 0.2 |
| | Number of training sequences | 200-1800 | 1800 | 1800 |

and the learning rate. The second category specifies the dataset configuration, including the quantity of training data sequences and the input and target time series window lengths.

In this study, each model architecture necessitates distinct hyperparameters, necessitating independent optimization for each.
The hyperparameters were optimized using a Bayesian optimization approach, leveraging the experiment manager functionality in MATLAB's statistics and machine learning toolbox MATLAB (2020). Due to computational limitations, the optimization is not carried out for each case and lidar pattern individually. Each model architecture has been optimized for case $A$ and $A_{Lidar}$ separately, and the optimized parameters have been applied to all cases. The hyperparameters, encompassing the parameter space and optimized values for both the time series and DEL prediction networks, are detailed in table 2. The Adaptive
Moment Estimation (ADAM) algorithm was employed to optimize the model parameters during training. Under the defined hyperparameters, training a single virtual load sensor model usually lasts approximately 1 hour, and predicting a ten-minute target sequence is accomplished in under 1 second.



## 3 Simulation setup

### 3.1 FOWT model

The University of Maine's VolturnUS-S reference floating wind turbine, detailed in Allen et al. (2020), features a semisubmersible substructure paired with the International Energy Agency's (IEA) 15-240-RWT 15 MW reference wind turbine, as described in Gaertner et al. (2014). This turbine has a rotor diameter of 240 meters and a hub height of 150 meters. The floating support structure is a steel design, comprising three radial and one central cylindrical body, with the turbine tower connected to the central column. Stability and station keeping are achieved through three mooring lines, each 850 meters long and the mass

per length is 685 kg/m, attached to the radial bodies. A visual representation of the floater, including its geometry, the locations of fairleads, and the lidar installation point, is depicted in Figure 5 (right). Comprehensive details on the design specifications of both the floater and the mooring lines are available in Allen et al. (2020). Additionally, to assist in the analysis of frequency contents in predicted time series, table 3 lists the natural frequencies of the FOWT model.

**Table 3.** Floater natural frequencies

| DOF | natural frequency |
|-------|-------------------|
| Surge | 0.007 Hz |
| Sway | 0.007 Hz |
| Heave | 0.049 Hz |
| Roll | 0.036 Hz |
| Pitch | 0.036 Hz |
| Yaw | 0.011 Hz |

### 3.2 Lidar Configuration

Four different lidar patterns, as depicted in figure 6, with an increasing number of focus points are investigated in this study. The figure shows the position of focus points in the y-z-plane (see figure 5) in front of the turbine. The measurements for each focus point are taken sequentially with a temporal distance of $1/n_{beam}$ seconds, where $n_{beam}$ is the number of focus points in the pattern. In this way, the time series of each LOS velocity has a sampling frequency of 1 Hz. The measurement range is set to 300m. The beam range gate length is set to 30 m and discretized by 10 equidistant points along the range gate. Additionally,

Gaussian white noise of $SNR = 20db$ is added to each measurement. This is done to mimic the uncertainties of real lidar measurements, which could originate from hardware components or data processing.





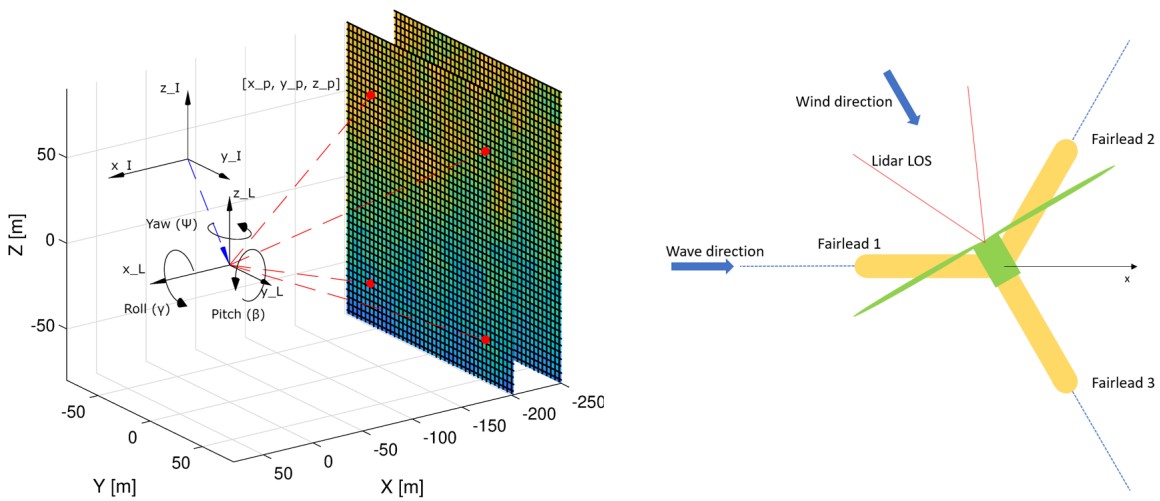

**Figure 5.** Left: Lidar pattern and coordinate systems for numerical lidar simulation. Right: Floater geometry and lidar position.

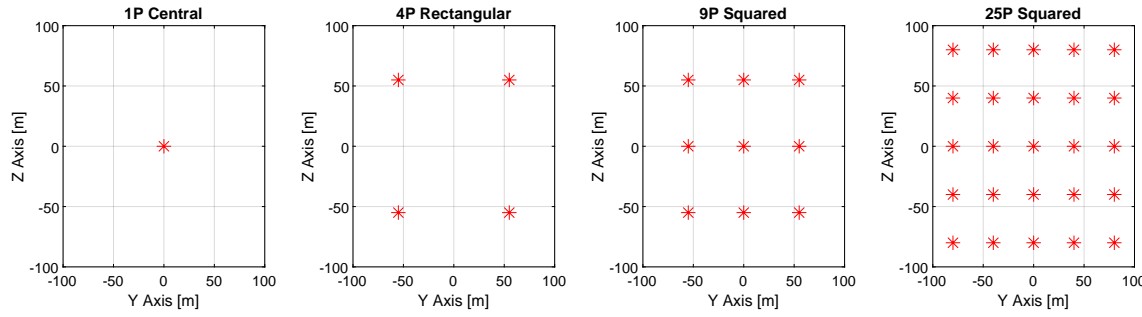

**Figure 6.** Lidar patterns considered in the analysis

### 3.3 Atmospheric conditions

A set of synthetic wind fields is generated using the Veers method for turbulence creation Veers et al. (1998). The total number of windfields is limited to 500 due to limited computational capacities. The mean wind speed of each wind field is randomized between 4ms$^{-1}$ and 20ms$^{-1}$, the turbulence intensity is randomized between 4% and 16%, and the power law shear exponent is randomized between 0 and 0.2. All wind field parameters are listed in table 4.

### 3.4 Creation of database

This study utilizes a dataset comprising 2200 simulations, each lasting 600 seconds, encompassing floater dynamics, lidar measurements, fairlead tensions, and DEL values. For each simulation, one turbulent wind field is randomly selected from



**Table 4.** Turbulent wind field parameters

| Parameter | Value |
|---|---|
| Wind speed [ms⁻¹] | 4:1:20 |
| Turbulence Intensity [%] | 4:1:16 |
| Surface Roughness [m] | 0.03 |
| Shear exponent [-] | 0.0:0.1:0.2 |
| Spatial Grid Resolution [m] | 5 |
| Grid size [m] | 300x250 |
| Timestep [s] | 0.05 |
| Usable Time [s] | 600 |

the aforementioned database of 500 wind fields. To mitigate the impact of transient effects typically present at the start of simulation runs, each simulation is extended to a total duration of 1200 seconds, with the initial 600 seconds being excluded from analysis. For the simulation of 1200 seconds turbine response, the periodic property of the generated turbulence boxes are used. Moreover, all sequences utilized in the training, validation, and testing phases are standardized, ensuring a mean of zero and a standard deviation of one across the entire dataset.

The ocean conditions in the aeroelastic simulations are determined based on the methodology proposed by Müller and Cheng (2018), which accounts for a basic correlation between wind speed, wave height, and wave period. To do this, wind speed is categorized into three distinct load ranges (LR): below-rated, rated, and above-rated. Each range has specified upper and lower limits for wave height and period. Depending on the mean wind speed of the chosen wind field, wave conditions are randomly sampled within these predefined boundaries. The specific parameters for wave conditions, corresponding to the different wind

speed ranges, are detailed in table 5.

**Table 5.** Parameter space wave conditions, where $H_s$ is the significant wave height and $T_p$ is the peak wave period.

| Parameter | LR1 [min, max] | LR2 [min, max] | LR3 [min, max] |
|---|---|---|---|
| Wind speed range [m/s] | 4:10 | 11:14 | 15:20 |
| $H_s$ [m] | 0.3, 3.2 | 0.5, 5.0 | 0.7, 7.0 |
| $T_p$ [s] | 1.7, 13.3 | 1.2, 12.3 | 0.9, 11.9 |

In each aeroelastic simulation conducted for this study, radial wind speed measurements are generated for four distinct lidar patterns. The wind direction, relative to a fixed Earth coordinate system, is randomly chosen within a range of 0 to 359 degrees, while the nacelle is always aligned with the wind direction. Similarly, the wave heading, also about the Earth-fixed system, is randomly selected across the same 0 to 359-degree range. To align with the 1 Hz sampling rate of the lidar radial wind



speed measurements, all feature and target time series within the simulations are downsampled accordingly. All predictions and performance evaluations for time series and DELs are only based on the downsampled time series. To develop the models, up to 1800 of these simulations are allocated for training. Additionally, 200 simulations are used for model validation during the hyperparameter optimization phase, and another 200 are reserved for the final testing of the model.

## 3.5   Measurement noise

In a pragmatic utilization context of the trained predictive models, the required input data has to be obtained from several sensors. Each distinct sensor category may be subject to measurement inaccuracies. To address the influence of noise present in the acquired measurement data, a noise modeling procedure has been implemented for every individual input feature of the models, subsequently introducing it into the simulated time series. It is assumed that an integrated GNSS/IMU sensor obtains the position and dynamics measurements, which combines an inertial measurement unit (IMU) with a global navigation satellite

system (GNSS). The IMU includes an accelerometer, a gyroscope, and a magnetometer. Based on this setup, a sensor fusion model is used to create measurements from the simulated ground truth dynamics and position signals. The noise of integrated sensor systems is dominated by white noise (see e.g. Blum and Dambeck (2020)). Therefore, the measurement noise of the output signals is modeled using a white noise process. The accuracy of output signals used in this study is based on fact sheets of commercially available GNSS/IMU receivers (see e.g. SBG-Systems (2023)). Accuracy parameters are summarized in table

290   6.

**Table 6.** Inertial navigation system accuracy values.

| Parameter | Value (RMS) |
|---|---|
| Pitch / Roll [deg] | 0.1 |
| Yaw [deg] | 0.2 |
| Position accuracy [m] | [1 1 0.05] |
| Velocity accuracy [m/s] | 0.05 |
| Angular velocity accuracy [deg/s] | 0.001 |
| Acceleration accuracy [m/s$^2$] | $2*10^{-4}$ |

## 4   Results

### 4.1   Time series prediction

Figure 7 shows the RMSEN values across the testing data set for all modeling cases and lidar patterns with and without the presence of noise in the dynamics measurements. Comparing cases $A$ to $E$ to the cases $A_{Lidar}$ to $F_{Lidar}$ (see table 1) for

all four lidar patterns, adding the LOS wind speed measurements from the lidar to the model features does not improve the




prediction accuracy. This indicates, that the use of lidar signals as predictors for fairlead tensions does not provide additional value compared to dynamics and SCADA signals.

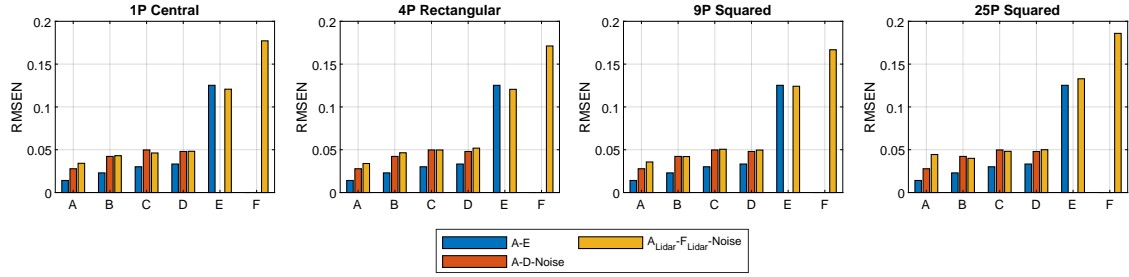

**Figure 7.** RMSEN of time series prediction for all cases and lidar patterns. Cases $A - D$-Noise as well as cases $A_{Lidar} - F_{Lidar}$-Noise denote cases with the influence of measurement noise in dynamics and position measurements.

The best performance is observed in case $A$, which incorporates all categories of platform dynamics, showing their collective importance in predicting fairlead tensions. A gradual increase in RMSEN is noted with the sequential removal of specific dynamics categories: rotational displacement, translational velocities, and then rotational velocities. Case D exhibits an RMSEN increase of approximately 0.015 compared to case A for noisy input signals. This relatively minor increase suggests that the translational position of the floater plays a pivotal role in time series signal prediction, with additional dynamics groups having less impact on the RMSEN. In case $F_{Lidar}$, all lidar patterns demonstrate elevated RMSEN values. A noteworthy trend is the reduction in RMSEN as the number of focus points in the lidar pattern increases, up to the 9P squared pattern. This implies that the ratio of informative content to the number of input features may be more favorable in the 9P pattern compared to the others. Regarding the impact of noise in the dynamics model inputs, a consistent pattern is seen across cases A to D, with RMSEN values rising by around 0.02. This uniform increase indicates that the effect of noise is independent of the type of input feature used and generally introduces an additional layer of uncertainty to the predictions.

Figure 8 illustrates a comparison between predicted time series and reference values, along with their Power Spectral Density plots, for cases $A$ through $E$. In case $A$, which includes both rotational and translational displacements and velocities, the model predicts successfully the time series with frequency components accurately up to 0.5 Hz. In case $B$, the predictions fail to capture frequency contents above 0.25 Hz, highlighting the significant role of rotational velocity inputs in the model's predictive capability. For cases $C$ and $D$, which lack data on both translational and rotational velocities, the model is limited to accurately reflecting only lower frequency fluctuations, up to approximately 0.15 Hz. Despite these variations in frequency content, it is important to note that the RMSEN differences between these cases are relatively small (as shown in figure 7). Therefore, the necessity and benefit of including additional input features largely depend on the specific application of the time series predictions. In contexts like the calculation of DELs, discussed in section 4.2, the ability of a model to account for higher frequency contents could significantly influence the resulting DEL values.

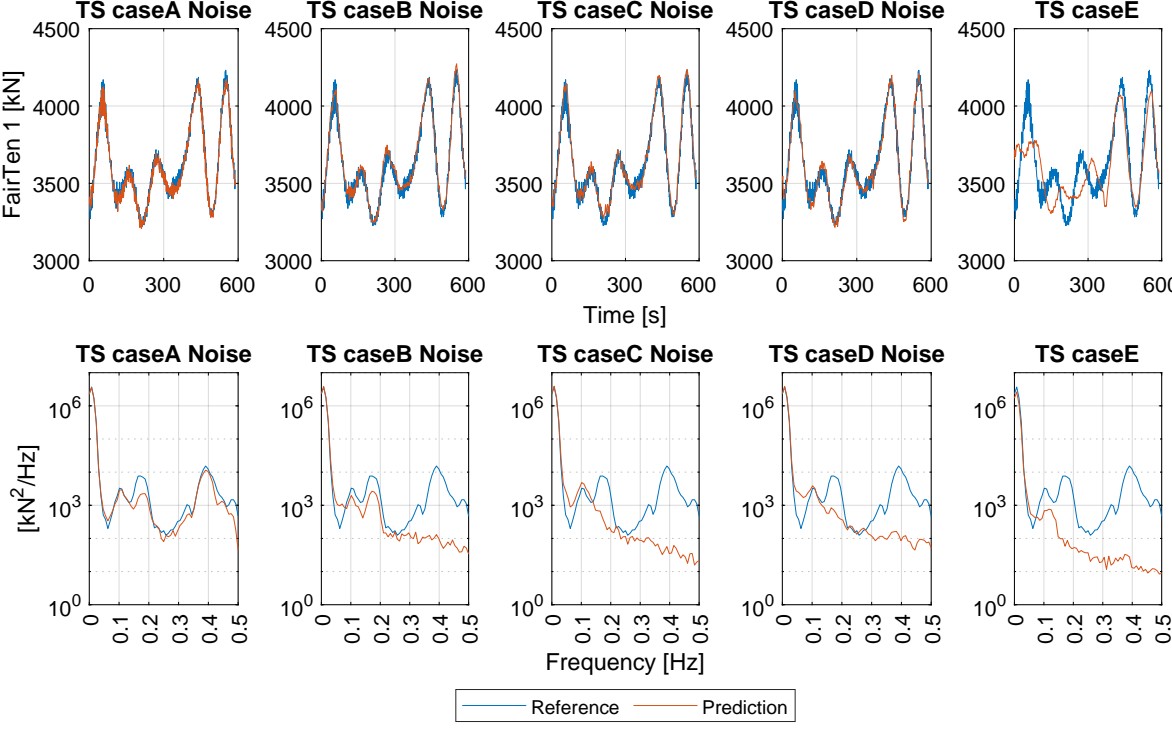

**Figure 8.** Example of time series predictions for cases $A - E$-Noise. Top: Prediction verus reference time series. Bottom: PSD of prediction vs. reference time series.

Figure 9 shows a time series prediction example of case $F_{Lidar}$ for lidar pattern 9P, which only uses the lidar LOS and nacelle yaw information as an input. This model can reproduce low-frequency fluctuations originating from variations in the wind field. Frequencies originating from the floater's response to the wave excitation and higher frequency dynamics of the floater are not captured by the model.





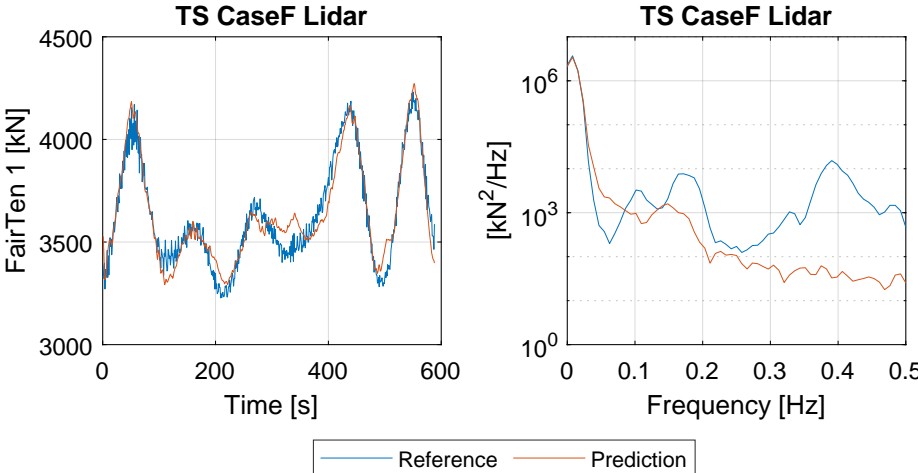

**Figure 9.** Example of time series predictions for case $F_{Lidar}$, 9P. Left: Prediction verus reference time series. Right: PSD of prediction vs. reference time series.

## 4.2 Fairlead tension DEL prediction

### 4.2.1 Prediction without lidar measurements

325 Figure 10 displays the prediction accuracy for DELs using three different model architectures, covering cases A through E. In each boxplot, the lower and upper bounds of the box represent the 25th and 75th percentiles of the data distribution, respectively. The median of the sample is indicated by the horizontal line situated in each box. Any data point marked as a circle is classified as an outlier; these are values that lie more than 1.5 times the interquartile range beyond the box's upper or lower boundaries. The whiskers of the boxplot extend from the edges of the box to the furthest data points that fall within the

330 range defined by the whisker length. The whisker range is defined through the highest/lowest values which do not fall in the definition of outliers.

Across different model architectures, several consistent trends are observed in this study. Case A uniformly outperforms other cases, suggesting that both rotational and translational displacements, along with their respective velocities, significantly contribute to the accuracy of DEL predictions. The similarity in the median error magnitude, error value distribution, and

335 outlier count across all models suggests that each modeling architecture is generally effective for DEL prediction tasks.

The gap in prediction accuracy between cases A and B is marginal for all models, hinting at a relatively minor role of rotational velocity inputs in overall DEL prediction accuracy. Conversely, cases C and D exhibit higher absolute percentage errors than A and B across all models, with this effect being more pronounced in the convolutional and Seq2One models. These cases also show a broader range of prediction errors and increased outliers. The difference in prediction accuracy between cases





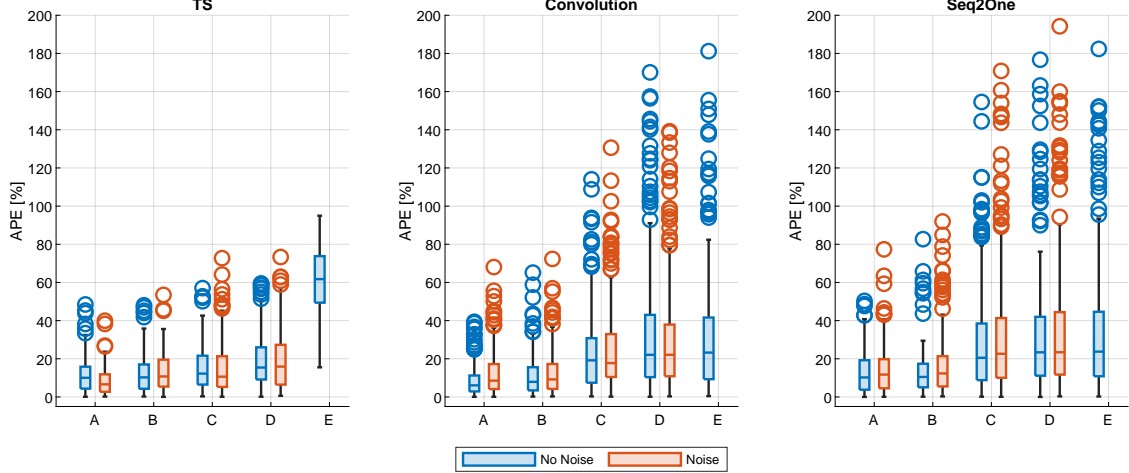

**Figure 10.** Absolute percentage erros of DELs for cases $A - E$ with and without influence of noise. Left: time series model. Middle: Convolutional model. Right: Sequence2One model.

C and D is relatively small for all models, indicating that rotational displacements have a smaller impact than translational displacements in predicting DELs.

Interestingly, the influence of noise on the input data is either minimal or inconsistent across all models, suggesting that the level of introduced noise does not significantly contribute to uncertainty in DEL predictions. In some instances, such as in case TS A, the presence of noise even slightly improves prediction accuracy. This could be attributed to enhanced generalization capabilities when models are trained on noisy data, a phenomenon that has been explored in existing literature (e.g., Um et al. (2017)).

Besides these similarities, the three models show significant differences in their ability to predict fairlead tension DELs. While the median error values for cases A and B are similar for all three models, cases C and D show approximately 10 % higher median errors, for the convolutional and Seq2One models.

The approach for calculating the DELs based on time series predictions shows a notably narrower range of error values, especially apparent in cases B, C, and D. Furthermore, both the convolutional and the seq2One models tend to predict a greater number of outliers, and the errors generated by these models span a wider range of magnitudes. This difference in performance can be attributed to the intrinsic characteristics of the different modeling approaches used. For the time series prediction model, DEL values are derived analytically. This method is robust in situations where the predicted time series are reasonably accurate in scale, and capture the low-frequency fluctuations of the signal. Under these conditions, the DEL calculation algorithm is less likely to yield DEL results with high errors. On the other hand, the convolutional and Seq2One models derive DEL predictions only on a statistical basis. They lack an inherent understanding of the analytical procedures of the DEL calculation. Consequently, as observed in this study, these models are more prone to higher prediction uncertainties.





Interestingly, the error median for case E is significantly higher for the time series model, compared to convolution and
seq2one, although the spread of error and the number of outliers is smaller. The prediction accuracy of the time series prediction
can explain this. As shown in figure 8, the time series model for case E cannot predict the low-frequency fluctuations of the
signal correctly. Consequently, the calculated DEL values show higher median errors.

Figure 11 shows the model predicted DELs about the reference DELs, calculated from the reference time series for all model
architectures and cases A to D. Additionally, the result of a linear regression and the coefficient of determination is shown for
each case. As indicated by figure 10 case A provides the best prediction accuracy for all models. However, for convolutional
and Seq2One models higher scatter around the 1-1 reference can be observed. Although the time series prediction shows the
least scatter around the 1-1 reference, a negative bias or underestimation of DELs predictions can be observed for the time
series approach. While this bias is relatively small for case A, it is more pronounced for cases with fewer model input features.
Particularly, large DEL values are strongly underestimated (See TS case E). The behavior of the time series prediction itself
can explain this. As shown in figure 8 the correct representation of the time series signals frequency content depends on the
availability of input features. Consequently, the calculated DEL values are negatively biased if the frequency content of the
reference signal is not reflected in the time series predictions.

The convolutional and Seq2One models exhibit different characteristics. A notable pattern is observed where both models
tend to overestimate smaller DEL values and underestimate larger ones. As a result, their predictions tend to be shifted towards
the center of the DEL distribution. This effect becomes more evident when the models have fewer input features. This trend
is especially prominent in cases C, D, and E, where the models' inability to predict smaller DEL values is observable. The
underlying reason for this effect possibly stems from the purely statistical nature of these models. Unlike the analytical DEL
calculations approach, the convolutional and Seq2One models cannot effectively distinguish between low-load conditions. In
contrast, the time series prediction model, combined with the analytical DEL calculation method, shows a better ability to
detect these subtle differences in low-load scenarios. This distinction in model performance highlights the limitations of a
purely statistical approach, particularly in contexts where distinguishing between closely related load conditions is crucial for
accurate DEL prediction.

Figures 12 and 13 show the relative error between predicted and reference DELs for two cases (Figure 12: TS Case A,
figure 13: Convolution Case A). The error is shown relative to the randomized environmental conditions used as input for the
aeroelastic simulations. The colors of individual data points reflect the absolute magnitude of the respective DEL prediction.

For TS Case A, the negative bias in DEL prediction is reflected by the distribution of relative Errors. It can also be observed,
that high relative errors occur for relatively small absolute DEL values. Conversely, data points with relatively large absolute
DEL values, show comparatively small relative errors. Concerning the sensitivity to environmental conditions, the relative error
depends on the wave conditions. High relative errors cumulate for low wave heights, indicating difficulties in predicting DEL
values correctly for low hydrodynamic loading scenarios. Similarly, large relative errors occur in low wind speeds indicating
inaccurate predictions for low aerodynamic loads and related turbine excitations. No clear influence of wind direction, wave
period, wave direction, and turbulence can be observed.



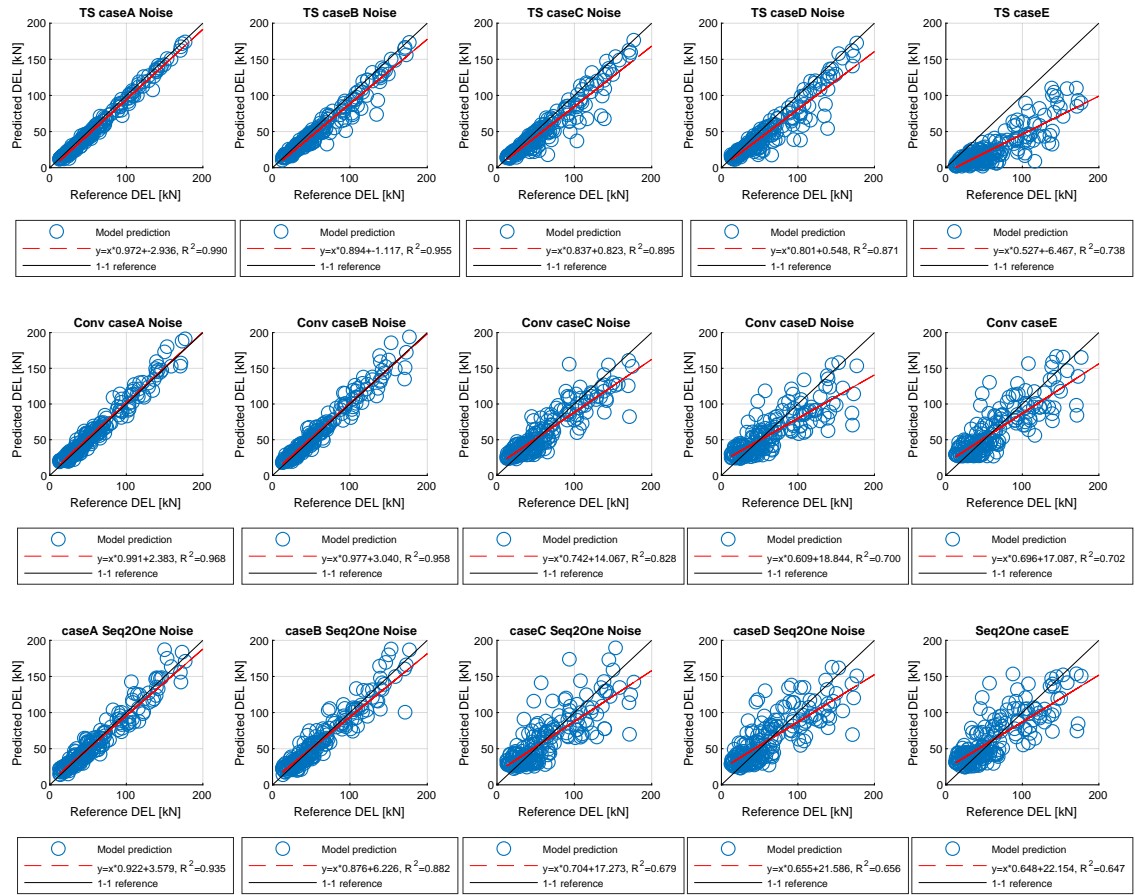

**Figure 11.** One-to-One comparison between predicted and reference reference DELs for cases $A - E$ under the influence of noise. Top: Time series model. Middle: Convolutional model. Bottom: Seq2One model.

For Case $A$, convolution (figure 13) different patterns can be observed. Overall increased scatter of DEL predictions is reflected in the relative errors. Similar to TS case A, large relative errors occur for low absolute DEL values. However large relative errors tend to be positive, indicating an overestimation of DELs. High relative errors tend to cumulate for low wave heights and low turbulence conditions. It is important to emphasize that the models are specifically trained to forecast absolute DEL values, focusing the training on reducing large absolute errors rather than relative ones. The relative error sensitivity in the Seq2One model displays consistent patterns with these observations, and as such, its results are not discussed separately.



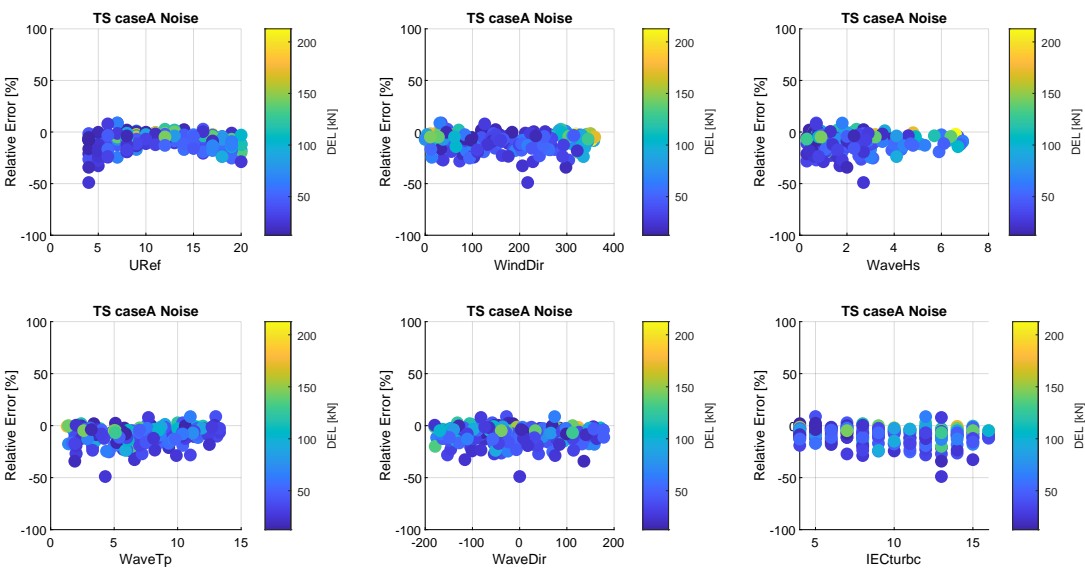

**Figure 12.** Sensitivity of relative Errors between predicted and reference DELs to environmental conditions for case $A$, time series model.

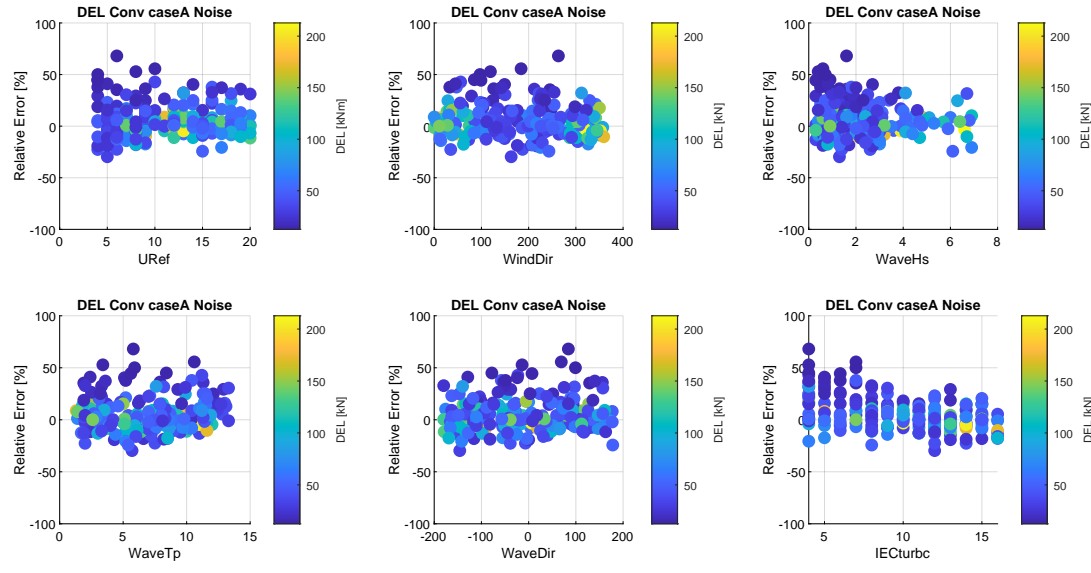

**Figure 13.** Sensitivity of relative Errors between predicted and reference DELs to environmental conditions for case $A$, Convolutional model.




### 4.3 Predictions using lidar measurements

In this section, the results of the DEL prediction using lidar inflow measurements are presented. The analysis of time series predictions for cases $A_{Lidar}$ to $E_{Lidar}$ (see section 4.1) revealed that the additional use of lidar measurements as model input features is not improving the prediction accuracy. This is also true for DEL prediction with all investigated model architectures and lidar patterns. Therefore, in the following, we focus on the discussion of predictions made only based on lidar measurements (case $F_{Lidar}$).

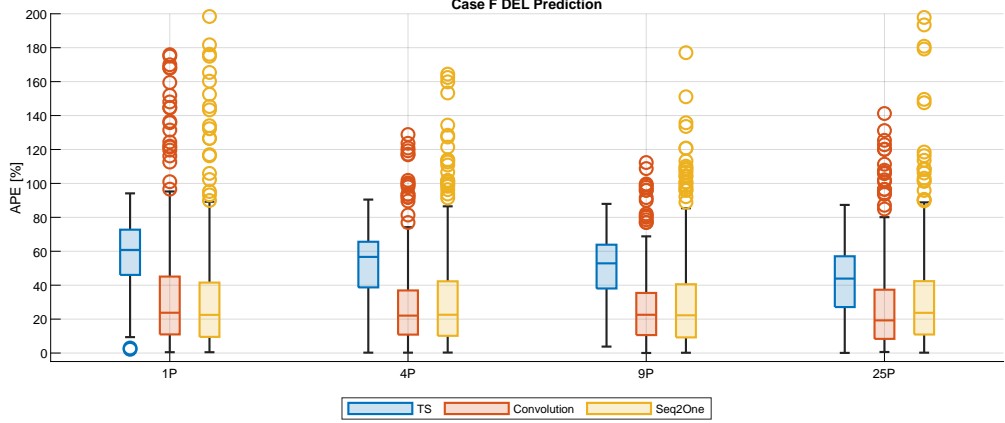

**Figure 14.** Absolute percentage errors of DELs for cases $F_{Lidar}$ for all lidar patterns and model architectures.

Figure 14 shows the absolute percentage error for case $F_{Lidar}$ for all three model architectures over the four investigated lidar patterns. The definition of the boxplots is the same as in the previous section. The comparison of results for different lidar patterns suggests a correlation between prediction accuracy and the density of points within the pattern. For time series predictions, there is a tendency for smaller errors in predictions for patterns with a higher number of points. For the time series prediction model, the most accurate predictions, as indicated by the median error, are observed for the 25-point (25P) pattern.
In contrast, for the convolutional and Seq2One models, the 9P pattern shows the best performance (lowest median error).

When comparing the overall performance, both the convolutional and Seq2One models demonstrate significantly lower median errors than the time series model. However, they also exhibit a higher incidence of outliers. This indicates that the time series model, DEL calculation procedure is more robust against high relative prediction errors. In this case, the median prediction accuracy is determined by the underlying prediction of fairlead tension time series based on lidar measurements.
Figure 15 shows predicted versus reference DELs for the 25P patterns and the three model architectures. The observed trends are similar to those observed for the non-lidar cases in the previous section. The time series model exhibits high negative bias, increasing with the absolute magnitude of DEL values. The Convolution and Seq2One models, tend to overestimate small DELs and underestimate large DELs. Particularly the inability of Seq2One and convolution models to predict small DEL values correctly can be observed.



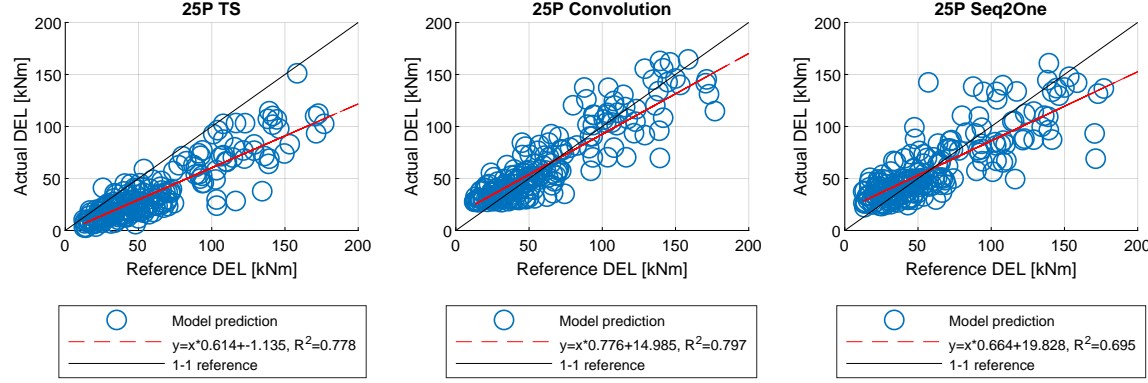

**Figure 15.** One-to-One comparison between predicted and reference reference DELs for cases $F_{Lidar}$. Left: Time series model. Middle: Convolutional model. Right: Seq2One model.

Figure 16 shows the sensitivity of prediction errors to the randomized environmental conditions in the input data for the convolutional model and 9P lidar pattern. While overall much higher relative error values and a wider range of error can be observed compared to the non-lidar cases, the visible trends are similar. High relative errors occur for small absolute DEL values while a positive bias for small DEL predictions and a negative bias for large DEL predictions is visible. No clear dependency of relative error values with wind speed, wind direction, and wave direction is visible. High relative errors occur

for low wave heights, indicating that the floater motion in these conditions cannot be extracted from measured lidar signals. Similarly high relative errors occur for low turbulence conditions.



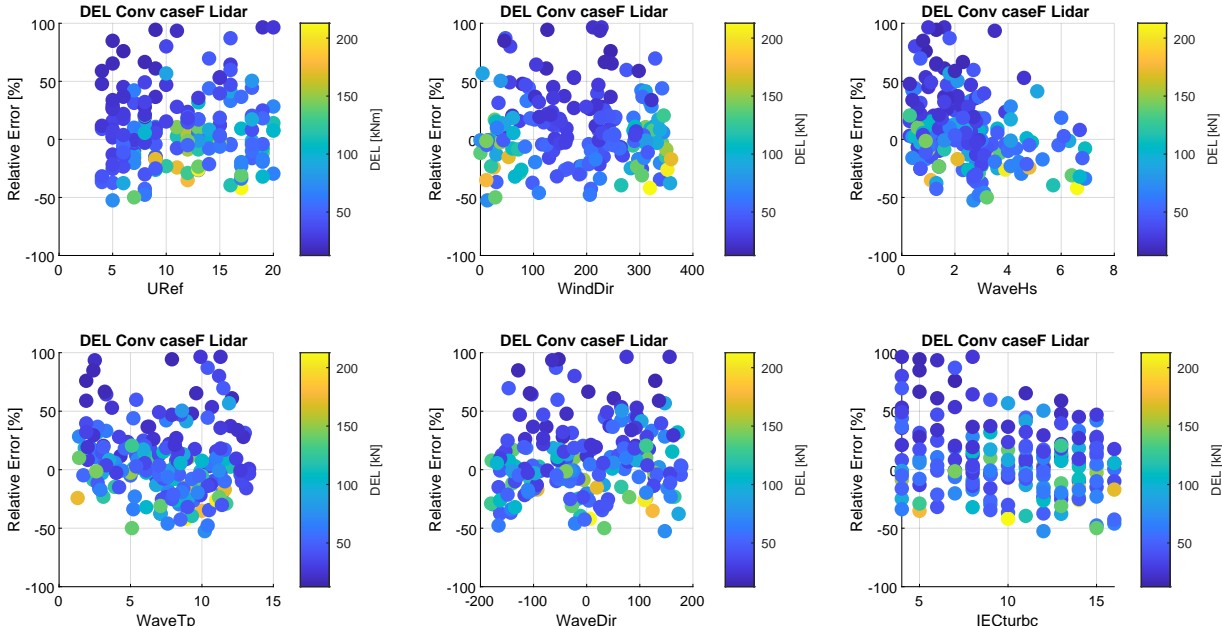

**Figure 16.** Sensitivity of relative Errors between predicted and reference DELs to environmental conditions for cases $F_{lidar}$, 9P, convolutional model.

## 5 Discussion

In the present study, different models for predicting fairlead tension time series and DEL have been developed. All models are trained to predict fairlead tension 1 (see figure 5). Due to the symmetry of the floater and the mooring line layout, it can be
expected, that the presented results represent all three fairleads. In the case of a non-symmetric floater design or mooring line layout individual models for each fairlead would need to be trained.

     The findings discussed here are derived from a specific floater model employed in this research. However, it can be expected that the core outcomes of this study do apply to other floater designs. This statementis based on the assumption that the dynamics of the platform fundamentally determine fairlead tension loads. In scenarios where these dynamics are accurately
captured through measurements, as in cases $A$ to $D$, it can be expected that training the model architectures presented in this study would yield similar results for different floater designs.

     The accuracy of predicting fairlead tension using nacelle-based lidar inflow measurements is likely to be significantly influenced by the specific dynamic behavior of the floater. As detailed in Gräfe et al. (2023), the impact of floater motions on nacelle-based lidar measurements is predominantly governed by the frequency and amplitude of movements in various degrees
of freedom, such as pitch or surge. Consequently, the effectiveness of lidar in capturing floater dynamics heavily depends on the design of the floater and the configuration of its mooring lines. In the case of the floater model used in this study, the effect of its





motion on lidar measurements is relatively minor. As a result, lidar data demonstrated limited efficacy in accurately predicting fairlead tensions. Furthermore, the analysis revealed only marginal differences among various lidar patterns, indicating that a simplistic representation of inflow, captured by a single time series signal, is adequate for achieving the observed results. This implies that similar measurements could potentially be obtained using simpler instruments like cup or sonic anemometers.

The data set used for model training, validation, and testing in this study is simulated using an aeroelastic wind turbine simulation code in combination with a numerical lidar simulator. While this simulation-based approach can show the general feasibility of the approach, several considerations have to be made for a real application. In a practical setting, the model training data could either be acquired through real measurements or a simulation model. If real fairlead tension measurement data is used, it must be ensured that it reflects all relevant environmental conditions. Especially in extreme conditions, gathering data in sufficient quantities could be difficult. The use of training data sets for other turbines of the same type, eg. in a fleet of FOWT, could be possible. However, even for FOWT identical in construction, differences in the orientation or the mooring line layout could lead to unexpected effects or bias in the load predictions. Suppose a FOWT simulation is used to generate the training database. In that case, it is essential that the simulation model is validated and reflects the behavior of the real system accurately across all relevant environmental conditions. Also, more sophisticated methods using a combination of simulated and measured data, as suggested by Schröder et al. (2022), e.g. through a transfer learning approach, could be beneficial in this context. All models in this study have been trained on the entire training data set, assuming no prior knowledge about environmental conditions in the prediction step. One approach to improve prediction accuracy could involve training separate models for specific environmental conditions e.g. bins of wind direction or wind speed. While this approach could potentially improve the performance of individual models it would require an additional process step in the prediction, selecting the specific model based on the present conditions.

In this study, sensor noise was modeled to represent model input data realistically. The analysis indicates that this modeled noise has a relatively small impact on time series predictions, though the exact magnitude of this influence is not specified. For the prediction of DELs, the effect of noise appears negligible, suggesting that other sources of uncertainty in the model play a more significant role than sensor noise. This finding underscores the potential practicality of using a virtual load sensor approach. However, it is important to note that the study only accounted for sensor noise. Other factors, such as systematic errors, sensor malfunctions, or uncorrected biases in measurements, were not considered and could lead to larger discrepancies in DEL predictions. In scenarios where such issues are present, supplementing predictions with lidar measurements could provide a valuable redundancy or serve as a reference to identify erroneous predictions. This is especially true, in case a lidar inflow measurement system is already available due to its use in the turbine's control system.

Three different model architectures were evaluated for the prediction of DELs. Among these, the procedure that calculates DELs from predicted time series, particularly for cases $A$ to $D$ that utilize floater dynamics as inputs, exhibited superior performance. This is true, in terms of both median errors and scatter of prediction errors. A possible explanation lies in the definition of the DEL metric. Variations in the time series are amplified to the power of the Wöhler exponent. Depending on the value of the Wöhler exponent, this leads to a higher realization-to-realization uncertainty compared to other statistical metrics such as the mean of a time series. While the time series prediction approach captures these uncertainties relatively well, the





purely statistical approaches show inferior performance in capturing these uncertainties. Future work should, therefore, include a sensitivity analysis, showing the effect of the Wöhler exponent on the prediction accuracy.

480 The convolutional network architecture yielded the best results for the predictions made based on lidar measurements. High relative errors were more frequent in instances of small absolute DEL values. This finding carries significance in scenarios where DELs are accumulated for long-term damage estimations, such as calculating a structure's remaining lifetime. In this case, the impact of high relative errors on overall damage is mitigated due to the nonlinear relationship between DEL and damage, depending on the specific value of the Wöhler coefficient. This aspect highlights the importance of understanding the error characteristics in DEL predictions, especially in their application to long-term structural health monitoring and life cycle 485 analysis.

The hyperparameters for each model were fine-tuned using a Bayesian optimization process, employing a distinct validation dataset for this purpose. However, due to constraints in computational resources and the extensive range of cases and pattern combinations, it was not feasible to optimize every scenario. Consequently, optimization was performed explicitly for cases $A$ and $A_{Lidar}$ across each model architecture. While the impact of optimized parameters on model performance was gener- 490 ally found to be modest, there remains a possibility that fine-tuning hyperparameters for all individual cases might enhance prediction accuracy in certain instances.

## 6 Conclusions

In this research, various machine learning approaches were explored for predicting fairlead tension time series and Damage Equivalent Loads (DELs), utilizing simulated dynamics and SCADA data from the UMaine VolturnUS-S reference Floating 495 Offshore Wind Turbine (FOWT). The study also examined the potential of incorporating inflow measurements from a forward-looking, nacelle-mounted lidar system, based on simulated lidar data. Additionally, the impact of noise on the dynamics input measurements was assessed by introducing realistic noise levels into the simulated signals.

For time series predictions, a Long Short-Term Memory neural network was employed. Our findings indicate that incorporating rotational and translational displacements and velocities achieves the highest prediction accuracy. The effect of noise in 500 the dynamics input measurements on prediction accuracy was found to be small (quantify the impact). Augmenting the input feature set with lidar measurements did not enhance prediction accuracy in the cases studied. Predictions based solely on lidar data broadly captured the reference sequences' order of magnitude and low-frequency fluctuations.

In predicting DELs, three methodologies were investigated. First, calculating DELs based on time series predictions; second, using a Sequence-to-One (Seq2One) LSTM network; and third, employing a convolutional neural network architecture. 505 Consistent with the time series predictions, including all examined platform dynamics inputs led to the highest accuracies across all model architectures. Adding lidar measurements to the dynamics input features did not improve accuracy. However, models relying solely on lidar measurements for predictions achieved comparable accuracies (with an Absolute Percentage Error (APE) of approximately 20%) to those utilizing platform displacement inputs. For predictions based solely on lidar data, direct DEL prediction by the neural network outperformed DEL calculation based on predicted time series.





Notably, measurement noise in the platform dynamics inputs had a negligible effect on our study's outcomes. Overall, our results suggest that measurements from a GNSS/INS sensor are crucial for accurately predicting fairlead tension time series and DELs. The integration of lidar inflow measurements could serve as a beneficial backup in cases of sensor failures or as a reference system to verify the proper functioning of the actual load prediction model.

*Code availability.*   The data that support the findings of this study are available from the corresponding author, MG, upon reasonable request.

*Author contributions.*   M.G. developed the virtual load sensor models and the software implementation. M.G. conducted the simulation studies and drafted the paper. V.P. contributed to the conceptualization, code review, and review of the paper. N.D. and P.W.C. contributed to discussions and reviewed the paper.

*Competing interests.*   The authors have the following competing interests: At least one of the (co-)authors is a member of the editorial board of Wind Energy Science.

*Acknowledgements.*   This study has received funding from the European Union's Horizon 2020 research and innovation programme under the Marie Skłodowska Curie grant agreement N° 860879.



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
