# Peer review of "Machine learning based virtual load sensors for mooring lines using motion and lidar measurements"

_Wind Energy Science, 2024_

## Referee Comment (RC2)

[referee-annotated manuscript omitted]

---

## Author Comment (AC1)

**Authors' Response Preprint wes 2024-25**

We thank all reviewers for their constructive comments and suggestions, which greatly improved the manuscript. We highly appreciate the time and effort the reviewers dedicated to this. In the following, we reply to the reviewers' comments point by point. Original comments are given in black, and answers are given in blue.

On behalf of the authors

Moritz Gräfe

**Reply to RC-1**

The authors would like to thank the anonymous reviewer for his constructive and useful feedback. All comments have been considered for the revised version of the manuscript. A list of individual comments and replies follows:

**General Comments**

The paper deals with a very interesting and innovative topic, which is the use of LiDAR and SCADA data for a model-based estimate of mooring lines damage equivalent loads. The research design is accurate and the simulations are extensively and clearly described. The results are well presented and coherent.

Thank you.

I understand that, given the innovativeness of the work, there are not so many references against which comparing. Anyway, I think that a deeper discussion about how good the results of Figure 10 are could improve the manuscript. Similarly, I am interested in further insights about how the results and the importance of the various features might change when dealing with measured, instead than simulated, data.

Discussion:

We agree with the reviewer that a deeper discussion on the quality of our results would be beneficial. However, the absence of reference studies investigating similar problems, coupled with the fact that our study only evaluates model accuracy quantitatively without considering the application of model results (e.g., in a monitoring system), makes this challenging. To provide the reader with a better understanding of the error ranges in our results, we have added the following sentence:

"Overall, the results presented in Figure 10 suggest that achieving DEL predictions with a median APE below 10% is feasible, provided the full input data set (Case A) is available, and the investigated modeling approaches are applied."

In our study, measurement noise is modeled in an attempt to better reflect the real characteristics of measurements. However, as discussed in Section 5, other sources of uncertainty might be more significant. This includes, e.g., Sensor malfunction, poor synchronization, loss of satellite signal, or sensor drift. These cannot be assessed using the tools and methods we use in our work. The discussion in section 5 has been updated, naming these uncertainties explicitly. With regards to the the importance of individual input

features when dealing with real measurements, our results reflect the descriptiveness of signals for predicting fairlead tensions. It can be expected these trends persist when using measured data, assuming the correct functioning of sensors and data acquisition systems.

Finally, the most important remark I have on the paper is that in my opinion it is fundamental to give the reader as soon as possible the information that the manuscript deals with simulated data. Therefore, I suggest to change the title in something like "Machine learning based virtual load sensors for mooring lines using simulated motion and lidar measurements".

We agree with the comment that the reader should be informed about the use of simulated data as soon as possible. Therefore, we have changed the title to " Machine learning-based virtual load sensors for mooring lines using simulated motion and lidar measurements." Additionally, we have changed the abstract (see line 3) to state the use of simulated data right in the beginning.

**Reply to RC-2**

The authors would like to thank the anonymous referee for the constructive and useful feedback. All comments have been considered for the revised version of the manuscript. A list of individual comments and replies follows:

**General comments**

The paper addresses core knowledge regarding influence of environmental conditions on mooring line loadings. The title is indeed misleading as suggested by another peer-review (e.g., Does it only estimate fairlead tension time series + DELs or also mooring line tensions). The relevant scientific questions are presented with specific details. The structure is well formed, with relevant figures and tables to describe the findings.

I did miss some validation steps, since the load sensors are virtual, the machine learning models have high uncertainty in themselves and using a virtual lidar is only able to replicate what we understand about the atmosphere. I have provided the relevant comments at specific locations, where I am missing the link or need more information and I hope the author is able to revise the manuscript to accommodate these requests.

Thank you for your helpful general feedback! Responses are given in the specific comments.

**Specific comments**

*Abstract*

Line 5: I am a bit confused with the goal of the paper. Does it only estimate fairlead tension time series + DELs or also mooring line tensions as written on line 11.

The paper estimates fairlead tension time series and DELs. Loads at other positions of the mooring line are not investigated separately. To avoid confusion about this, the wording has been changed in several instances in the paper. See lines 5, 11, and 63.

Line 10: Since the methodology first uses nacelle mounted lidar, predicts floater motion and wind speed based on some assumptions and then predicts the fairlead time series and DELS, it would make more sense to include a validation step for the model, e.g. to get the tower loads or blade loads which are more or less known.

We acknowledge that the lidar-based prediction of other signals, such as tower or blade loads, would be an interesting field for further investigation and should be considered in future work. However, since the characteristics of these loads are different from fairlead tensions, we believe that a machine learning model's ability to predict these loads could not directly validate a fairlead tension prediction model.

Line 22: use proper citation style here. either in brackets or building it in the sentence as "GWEC (2022) projections indicate..."

Corrected.

Line 26: which consequences? Either mention the consequences or mark a reference where they are addressed.

The sentence has been changed to: "Specifically, the mooring lines of FOWTs, which are vital for stability, are susceptible to mechanical failures that could lead to severe consequences, including safety hazards, environmental damage, and economic losses."

Line 29: the remaining lifespan of individual lines

Changed.

Line 53: Can you compare the mooring line tension forecasts with your prediction or state the uncertainties in forecasts between the two models (Walker's and yours). I would like to understand how far are you from one another in terms of forecasting accuracy.

It is difficult to directly compare the results of the two studies. Walker et al.'s work is focused on time series prediction and forecasting. Our work is focused on predicting DELs using time series predictions as an intermediate step, while no forecasts are made. Walker et al. report accuracy using absolute metrics, which cannot be quantitatively compared to the RMSEN metric used in our study. It should also be noted that the two studies consider different floater concepts (semisubmersible and spar), which show different dynamic behaviors. As discussed in section 2.2 of our paper, this does significantly influence the fairlead tension characteristics. For all the previous reasons a direct comparison cannot be made between the two models in terms of loads prediction.

Line 58: This is a complicated sentence. Could you simplify the sentence.

The sentence has been changed to: Various studies have used Nacelle-based lidar inflow measurements for turbine load calculation. For instance, in Dimitrov et al. (2019) and Conti et al. (2021), measured lidar wind speed time series are used to constrain synthetic turbulent wind fields and lidar estimated wind field statistics are used to parameterize the constrained wind fields.

Line 72: I am missing a figure, which details a mooring system and/or it's elements, a FOWT with a nacelle lidar with some information like DOFs

Section 3 (simulation setup) introduces the used turbine model, mooring line layout, and lidar characteristics. This section includes a figure showing the details requested by the reviewer. However, for clarity and readability, we believe it is better not to introduce the figure in the introduction section of the paper.

Figure 1: I am missing a feedback loop there from performance evaluation to the model tuning. Hopefully, you have done some feedback tuning here.

The created database is divided into three subsets of data. First, the "training" data set is only used to train the model. Second, the validation data set is used to tune the hyperparameters of the models. Here, a feedback loop between model accuracy and the chosen hyperparameters is used to find the optimal set of hyperparameters. Third, the testing data set is only used for the final performance evaluation. Here, no feedback loop exists for the model hyperparameters. This separation between validation and testing data sets is implemented to avoid model biases that could occur when tuning hyperparameters based on testing accuracy. To represent this approach, we have altered Figure 1, showing a feedback loop between validation and model tuning.

Line 37: are there other examples where such an application exists (other offshore structures or aviation)

Many other applications of virtual sensor models exist (e.g., automotive applications, bearings, gearboxes, etc.), mostly independent from specific industries but more related to the physical characteristics of the asset and the available data. To avoid confusion of the reader about the scope of the paper, we prefer not to discuss further applications at this point. The sentence has been changed to clarify that the modeling task investigated in this work falls into the category suitable for data-driven approaches.

"This is the case for predicting fairlead tensions from platform positions, dynamics, and lidar inflow measurements, as investigated in this work."

Line 165: in my opinion, setting this number to 600 is very small and can introduce errors. Usually, the number is used in the wind industry for 20 years times the frequency i.e. 20 years in 10 min cycles or 1 second cylces.

We acknowledge the relevance of the number of reference cycles in cases where damage is accumulated over extended periods of time and remaining lifetime calculations. In our study, as suggested by the reviewer, 1Hz DELs in 10-minute windows are used. However, DELs are only compared relative to each other. Therefore, in our opinion, the chosen number of reference cycles does not introduce errors. If the reviewer means that the chosen 10-minute time window is too short to capture low-frequency load cycles, we agree with the statement. However, we believe this is not critical for our study, which is not about the correct determination of DELs but about assessing the model's ability to predict those DELs. Here, it should also be noted that the machine learning models do not necessarily need full dynamics

cycles in the input data to predict the target time series correctly, as they only rely on patterns in the input data and have no physical "understanding" of frequency contents. In our case, time series predictions are made by dividing the 10-minute windows into smaller subsequences and concatenating the prediction afterward.

Line 192: If you have reduced the calculation of DELs to 600s , why not have the RMSEN value for each of those 600s? In that way, you will also be able to find out which situations are suitable for the model and which situations fail.

We agree with the statement that providing the RMSEN for each individual sample (or distribution across different situations) would provide additional information. Since time series are used as an intermediate step for DEL calculation in our study, we have chosen to show the time series prediction accuracy in terms of an aggregated metric, which gives the reader an overview of the differences between the cases. The distribution of errors is given for DELs (see. Figure 10) and also the error sensitivity to environmental conditions (see figures 12, 13, 16). This includes implicitly also the time series predictions as they are used for DEL calculation.

Table 2: How sensitive are the models and results to these hyperparameter pertubations. How stable are they for the different DLCs in wind turbine loading and different environment conditions?

The sensitivity of model results to hyperparameter perturbations has not been investigated. Hyperparameters are tuned using the validation data set, while results are only presented for the tuned (optimized) set of hyperparameters. In our study, we did not consider DLCs as the goal of our study was to build models that can predict fairlead loads under a wide range of normal operating conditions. This allows for the implementation of general models without the need for different models trained for specific load cases. We acknowledge that the inclusion of extreme events, start-ups, etc., would be an interesting extension for our work and will consider this suggestion in future work.

Table 4: Can you also provide the mean and standard deviations of input wind field parameters and that of output windfield parameters?

Table 4 shows the input parameters space from which the parameters of each individual wind field's parameters are sampled. 4:1:20 denotes that the wind speed is sampled from a uniform distribution between 4 and 20 m/s. Similarly, the parameter space for the turbulence intensity and the shear exponent is defined. If the reviewer refers to the fact that the actual statistical properties of the generated wind fields may slightly differ from the input parameters, we state that this effect has not been investigated. The developed prediction models only use generated time series from the aeroelastic/lidar simulation as inputs. Therefore, we believe that small deviations in the wind field statistics do not affect the prediction quality of the models.

We hope this response clarifies the reviewer's question. If we have misunderstood the reviewer's comment, please feel free to contact us again.

Line 303: It is noteworthy, but the reduction is very marginal in my opinion.

We agree with the comment and have changed the sentence to: "A noteworthy but minor trend is the reduction in RMSEN as the number of focus points in the lidar pattern increases up to the 9P squared pattern."

Figure 8:I observe two peaks in the PSD and I am wondering if you did some frequency and vibrational analysis to detect where these frequencies originate from?

An analysis of the signal has been conducted. In the example shown in the time series prediction, the first peak (around 0,15 Hz) corresponds to the peak wave frequency. The second peak (around 0,4) corresponds to the coupled structural dynamics of the floater/tower assembly. It should be noted that the frequency content observed in the predicted/target time is dependent on the environmental conditions. We have added this information in the revised manuscript (see line 312).

Figure 9: It would be interesting to understand where do the two peaks originate from?

The figure shows the same example of target time series. Please refer to the previous comment.

Figure 10: Why are there cases where the noise results are better than the ones without noise?

This observation is discussed in line 352.

Line 345: one reference is not a literature. Either mention just the name canceling the literature, or mention the whole list of literature references (or a couple of famous ones).

We agree with the reviewer's comment and have altered the sentence to: This could be attributed to enhanced generalization capabilities when models are trained on noisy data (see eg. Um et al. (2017)).